# Don't Blame the ELBO!
# A Linear VAE Perspective on Posterior Collapse

**James Lucas**[‡][*], **George Tucker**[†], **Roger Grosse**[‡], **Mohammad Norouzi**[†]

[‡]University of Toronto          [†]Google Brain

## Abstract

Posterior collapse in Variational Autoencoders (VAEs) arises when the variational posterior distribution closely matches the prior for a subset of latent variables. This paper presents a simple and intuitive explanation for posterior collapse through the analysis of linear VAEs and their direct correspondence with Probabilistic PCA (pPCA). We explain how posterior collapse may occur in pPCA due to local maxima in the log marginal likelihood. Unexpectedly, we prove that the ELBO objective for the linear VAE does not introduce additional spurious local maxima relative to log marginal likelihood. We show further that training a linear VAE with exact variational inference recovers an identifiable global maximum corresponding to the principal component directions. Empirically, we find that our linear analysis is predictive even for high-capacity, non-linear VAEs and helps explain the relationship between the observation noise, local maxima, and posterior collapse in deep Gaussian VAEs.

## 1 Introduction

The generative process of a deep latent variable model entails drawing a number of latent factors from the prior and using a neural network to convert such factors to real data points. Maximum likelihood estimation of the parameters requires marginalizing out the latent factors, which is intractable for deep latent variable models. The influential work of Kingma and Welling [24] and Rezende et al. [35] on Variational Autoencoders (VAEs) enables optimization of a tractable lower bound on the likelihood via a reparameterization of the Evidence Lower Bound (ELBO) [21, 6]. This has led to a surge of recent interest in automatic discovery of the latent factors of variation for a data distribution based on VAEs and principled probabilistic modeling [18, 7, 10, 16].

Unfortunately, the quality and the number of the latent factors learned is influenced by a phenomenon known as *posterior collapse*, where the generative model learns to ignore a subset of the latent variables. Most existing papers suggest that posterior collapse is caused by the KL-divergence term in the ELBO objective, which directly encourages the variational distribution to match the prior [7, 25, 38]. Thus, a wide range of heuristic approaches in the literature have attempted to diminish the effect of the KL term in the ELBO to alleviate posterior collapse [7, 33, 38, 20]. While holding the KL term responsible for posterior collapse makes intuitive sense, the mathematical mechanism of this phenomenon is not well understood. In this paper, we investigate the connection between posterior collapse and spurious local maxima in the ELBO objective through the analysis of linear VAEs. Unexpectedly, we show that spurious local maxima may arise even in the optimization of exact marginal likelihood, and such local maxima are linked with a collapsed posterior.

While linear autoencoders [37] have been studied extensively [4, 26], little attention has been given to their variational counterpart from a theoretical standpoint. A well-known relationship exists between linear autoencoders and PCA – the optimal solution of a linear autoencoder has decoder weight

---

[*]Intern at Google Brain
   Code available at https://sites.google.com/view/dont-blame-the-elbo

columns that span the same subspace as the one defined by the principal components [4]. Similarly, the maximum likelihood solution of probabilistic PCA (pPCA) [39] recovers the subspace of principal components. In this work, we show that a linear variational autoencoder can recover the solution of pPCA. In particular, by specifying a diagonal covariance structure on the variational distribution, one can recover an identifiable autoencoder, which at the global maximum of the ELBO recovers the exact principal components as the columns of the decoder's weights. Importantly, we show that the ELBO objective for a linear VAE does not introduce any local maxima beyond the log marginal likelihood.

The study of linear VAEs gives us new insights into the cause of posterior collapse and the difficulty of VAE optimization more generally. Following the analysis of Tipping and Bishop [39], we characterize the stationary points of pPCA and show that the *variance of the observation model* directly influences the stability of local stationary points corresponding to posterior collapse – it is only possible to escape these sub-optimal solutions by simultaneously reducing noise and learning better features. Our contributions include:

- We verify that linear VAEs can recover the true posterior of pPCA. Further, we prove that the global optimum of the linear VAE recovers the principal components (not just their spanning sub-space). More importantly, we prove that using ELBO to train linear VAEs does not introduce any additional spurious local maxima relative to log marginal likelihood training.

- While high-capacity decoders are often blamed for posterior collapse, we show that posterior collapse may occur when optimizing log marginal likelihood even without powerful decoders. Our experiments verify the analysis of the linear setting and show that these insights extend even to high-capacity non-linear VAEs. Specifically, we provide evidence that the observation noise in deep Gaussian VAEs plays a crucial role in overcoming local maxima corresponding to posterior collapse.

## 2 Preliminaries

**Probabilistic PCA.** The probabilitic PCA (pPCA) model is defined as follows. Suppose latent variables $\mathbf{z} \in \mathbb{R}^k$ generate data $\mathbf{x} \in \mathbb{R}^n$. A standard Gaussian prior is used for $\mathbf{z}$ and a linear generative model with a spherical Gaussian observation model for $\mathbf{x}$:

$$
\begin{aligned}
p(\mathbf{z}) &= \mathcal{N}(\mathbf{0}, \mathbf{I}) \,, \\
p(\mathbf{x} \mid \mathbf{z}) &= \mathcal{N}(\mathbf{W}\mathbf{z} + \boldsymbol{\mu}, \sigma^2 \mathbf{I}) \,.
\end{aligned}
\tag{1}
$$

The pPCA model is a special case of factor analysis [5], which uses a spherical covariance $\sigma^2 \mathbf{I}$ instead of a full covariance matrix. As pPCA is fully Gaussian, both the marginal distribution for $\mathbf{x}$ and the posterior $p(\mathbf{z} \mid \mathbf{x})$ are Gaussian, and unlike factor analysis, the maximum likelihood estimates of $\mathbf{W}$ and $\sigma^2$ are tractable [39].

**Variational Autoencoders.** Recently, amortized variational inference has gained popularity as a means to learn complicated latent variable models. In these models, the log marginal likelihood, $\log p(\mathbf{x})$, is intractable but a variational distribution, denoted $q(\mathbf{z} \mid \mathbf{x})$, is used to approximate the posterior $p(\mathbf{z} \mid \mathbf{x})$, allowing tractable approximate inference using the Evidence Lower Bound (ELBO):

$$
\begin{aligned}
\log p(\mathbf{x}) &= \mathbb{E}_{q(\mathbf{z}|\mathbf{x})}[\log p(\mathbf{x}, \mathbf{z}) - \log q(\mathbf{z} \mid \mathbf{x})] + D_{KL}(q(\mathbf{z} \mid \mathbf{x})||p(\mathbf{z} \mid \mathbf{x})) & (2) \\
&\geq \mathbb{E}_{q(\mathbf{z}|\mathbf{x})}[\log p(\mathbf{x}, \mathbf{z}) - \log q(\mathbf{z} \mid \mathbf{x})] & (3) \\
&= \mathbb{E}_{q(\mathbf{z}|\mathbf{x})}[\log p(\mathbf{x} \mid \mathbf{z})] - D_{KL}(q(\mathbf{z} \mid \mathbf{x})||p(\mathbf{z})) \qquad (:= ELBO) & (4)
\end{aligned}
$$

The ELBO [21, 6] consists of two terms, the KL divergence between the variational distribution, $q(\mathbf{z}|\mathbf{x})$, and prior, $p(\mathbf{z})$, and the expected conditional log-likelihood. The KL divergence forces the variational distribution towards the prior and so has reasonably been the focus of many attempts to alleviate posterior collapse. We hypothesize that the log marginal likelihood itself often encourages posterior collapse.

In Variational Autoencoders (VAEs), two neural networks are used to parameterize $q_\phi(\mathbf{z}|\mathbf{x})$ and $p_\theta(\mathbf{x}|\mathbf{z})$, where $\phi$ and $\theta$ denote two sets of neural network weights. The encoder maps an input $\mathbf{x}$ to the parameters of the variational distribution, and then the decoder maps a sample from the variational distribution back to the inputs.

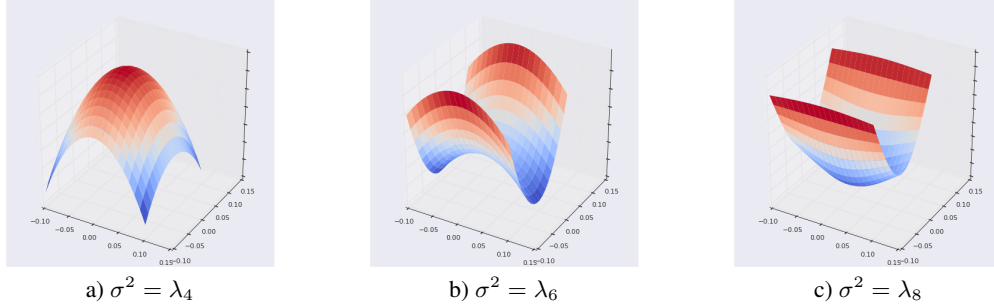

a) $\sigma^2 = \lambda_4$　　　　　　　　b) $\sigma^2 = \lambda_6$　　　　　　　　c) $\sigma^2 = \lambda_8$

Figure 1: **Stationary points of pPCA.** Two zero-columns of $\mathbf{W}$ are perturbed in the directions of two orthogonal principal components ($\mu_5$ and $\mu_7$) and the optimization landscape around zero-columns is shown, where the goal is to maximize log marginal likelihood. The stability of the stationary points depends critically on $\sigma^2$ (the observation noise). Left: $\sigma^2$ is too large to capture either principal component. Middle: $\sigma^2$ is too large to capture one of the principal components. Right: $\sigma^2$ is able to capture both principal components.

**Posterior collapse.** A dominant issue with VAE optimization is posterior collapse, in which the learned variational distribution is close to the prior. This reduces the capacity of the generative model, making it impossible for the decoder network to make use of the information content of all of the latent dimensions. While posterior collapse is widely acknowledged, formally defining it has remained a challenge. We introduce a formal definition in Section 6.2 which we use to measure posterior collapse in trained deep neural networks.

## 3 Related Work

Dai et al. [14] discuss the relationship between robust PCA methods [8] and VAEs. They show that at stationary points the VAE objective locally aligns with pPCA under certain assumptions. We study the pPCA objective explicitly and show a direct correspondence with linear VAEs. Dai et al. [14] showed that the covariance structure of the variational distribution may smooth out the loss landscape. This is an interesting result whose interactions with ours is an exciting direction for future research.

He et al. [17] motivate posterior collapse through an investigation of the learning dynamics of deep VAEs. They suggest that posterior collapse is caused by the inference network lagging behind the true posterior during the early stages of training. A related line of research studies issues arising from approximate inference causing a mismatch between the variational distribution and true posterior [12, 22, 19]. By contrast, we show that posterior collapse may exist even when the variational distribution matches the true posterior exactly.

Alemi et al. [2] used an information theoretic framework to study the representational properties of VAEs. They show that with infinite model capacity there are solutions with equal ELBO and log marginal likelihood which span a range of representations, including posterior collapse. We find that even with weak (linear) decoders, posterior collapse may occur. Moreover, we show that in the linear case this posterior collapse is due entirely to the log marginal likelihood.

The most common approach for dealing with posterior collapse is to anneal a weight on the KL term during training from 0 to 1 [7, 38, 30, 18, 20]. Unfortunately, this means that during the annealing process, one is no longer optimizing a bound on the log-likelihood. Also, it is difficult to design these annealing schedules and we have found that once regular ELBO training resumes the posterior will typically collapse again (Section 6.2).

Kingma et al. [25] propose a constraint on the KL term, termed "free-bits", where the gradient of the KL term per dimension is ignored if the KL is below a given threshold. Unfortunately, this method reportedly has some negative effects on training stability [33, 11]. Delta-VAEs [33] instead choose prior and variational distributions such that the variational distribution can never exactly recover the prior, allocating free-bits implicitly. Several other papers have studied alternative formulations of the VAE objective [34, 13, 2, 29, 41]. Dai and Wipf [13] analyzed the VAE objective to improve image fidelity under Gaussian observation models and also discuss the importance of the observation noise. Other approaches have explored changing the VAE network architecture to help alleviate posterior collapse; for example adding skip connections [30, 15]

Rolinek et al. [36] observed that the diagonal covariance used in the variational distribution of VAEs encourages orthogonal representations. They use linearizations of deep networks to prove their results under a modification of the objective function by explicitly ignoring latent dimensions with posterior collapse. Our formulation is distinct in focusing on linear VAEs without modifying the objective function and proving an exact correspondence between the global solution of linear VAEs and the principal components.

Kunin et al. [26] studied the optimization challenges in the linear autoencoder setting. They exposed an equivalence between pPCA and Bayesian autoencoders and point out that when $\sigma^2$ is too large information about the latent code is lost. A similar phenomenon is discussed in the supervised learning setting by Chechik et al. [9]. Kunin et al. [26] also showed that suitable regularization allows the linear autoencoder to recover the principal components up to rotations. We show that linear VAEs with a diagonal covariance structure recover the principal components *exactly*.

## 4   Analysis of linear VAE

This section compares and analyzes the loss landscapes of both pPCA and linear variational autoencoders. We first discuss the stationary points of pPCA and then show that a simple linear VAE can recover the global optimum of pPCA. Moreover, when the data covariance eigenvalues are distinct, the linear VAE identifies the individual principal components, unlike pPCA, which recovers only the PCA subspace. Finally, we prove that ELBO does not introduce any additional spurious maxima to the loss landscape.

### 4.1   Probabilistic PCA Revisited

The pPCA model (Eq. (1)) is a fully Gaussian linear model, thus we can compute both the marginal distribution for $\mathbf{x}$ and the posterior $p(\mathbf{z} \mid \mathbf{x})$ in closed form:

$$p(\mathbf{x}) = \mathcal{N}(\boldsymbol{\mu}, \mathbf{W}\mathbf{W}^\top + \sigma^2 \mathbf{I}), \tag{5}$$

$$p(\mathbf{z} \mid \mathbf{x}) = \mathcal{N}(\mathbf{M}^{-1}\mathbf{W}^\top(\mathbf{x} - \boldsymbol{\mu}), \sigma^2 \mathbf{M}^{-1}), \tag{6}$$

where $\mathbf{M} = \mathbf{W}^\top\mathbf{W} + \sigma^2 \mathbf{I}$. This model is particularly interesting to analyze in the setting of variational inference, as the ELBO can also be computed in closed form (see Appendix C).

**Stationary points of pPCA**   We now characterize the stationary points of pPCA, largely repeating the thorough analysis of Tipping and Bishop [39] (see Appendix A of their paper). The maximum likelihood estimate of $\boldsymbol{\mu}$ is the mean of the data. We can compute $\mathbf{W}_{\mathrm{MLE}}$ and $\sigma^2_{\mathrm{MLE}}$ as follows:

$$\sigma^2_{\mathrm{MLE}} = \frac{1}{n-k} \sum_{j=k+1}^{n} \lambda_j, \tag{7}$$

$$\mathbf{W}_{\mathrm{MLE}} = \mathbf{U}_k (\boldsymbol{\Lambda}_k - \sigma^2_{\mathrm{MLE}} \mathbf{I})^{1/2} \mathbf{R}. \tag{8}$$

Here $\mathbf{U}_k$ corresponds to the first $k$ principal components of the data with the corresponding eigenvalues $\lambda_1, \ldots, \lambda_k$ stored in the $k \times k$ diagonal matrix $\boldsymbol{\Lambda}_k$. The matrix $\mathbf{R}$ is an arbitrary rotation matrix which accounts for weak identifiability in the model. We can interpret $\sigma^2_{MLE}$ as the average variance lost in the projection. The MLE solution is the global optimum. Other stationary points correspond to zeroing out columns of $\mathbf{W}_{\mathrm{MLE}}$ (posterior collapse).

**Stability of $\mathbf{W}_{\mathrm{MLE}}$**   In this section we consider $\sigma^2$ to be fixed and not necessarily equal to the MLE solution. Equation 8 remains a stationary point when the general $\sigma^2$ is swapped in. One surprising observation is that $\sigma^2$ directly controls the stability of the stationary points of the log marginal likelihood (see Appendix A). In Figure 1, we illustrate one such stationary point of pPCA for different values of $\sigma^2$. We computed this stationary point by taking $\mathbf{W}$ to have three principal component columns and zeros elsewhere. Each plot shows the same stationary point perturbed by two orthogonal vectors corresponding to other principal components.

The stability of the pPCA stationary points depends on the size of $\sigma^2$ — as $\sigma^2$ increases the stationary point tends towards a stable local maximum so that we cannot learn the additional components. Intuitively, the model prefers to explain deviations in the data with the larger observation noise. Fortunately, decreasing $\sigma^2$ will increase likelihood at these stationary points so that when learning $\sigma^2$ simultaneously these stationary points are saddle points [39]. Therefore, learning $\sigma^2$ is necessary for gaining a full latent representation.

## 4.2 Linear VAEs recover pPCA

We now show that linear VAEs can recover the globally optimal solution to Probabilistic PCA. We will consider the following VAE model,

$$
\begin{aligned}
p(\mathbf{x} \mid \mathbf{z}) &= \mathcal{N}(\mathbf{W}\mathbf{z} + \boldsymbol{\mu}, \sigma^2 \mathbf{I}), \\
q(\mathbf{z} \mid \mathbf{x}) &= \mathcal{N}(\mathbf{V}(\mathbf{x} - \boldsymbol{\mu}), \mathbf{D}),
\end{aligned}
\tag{9}
$$

where $\mathbf{D}$ is a diagonal covariance matrix, used globally for all of the data points. While this is a significant restriction compared to typical VAE architectures, which define an amortized variance for each input point, this is sufficient to recover the global optimum of the probabilistic model.

**Lemma 1.** *The global maximum of the ELBO objective (Eq. (4)) for the linear VAE (Eq. (9)) is identical to the global maximum for the log marginal likelihood of pPCA (Eq. (5)).*

*Proof.* Note that the global optimum of pPCA is defined up to an orthogonal transformation of the columns of $\mathbf{W}$, *i.e.,* any rotation $\mathbf{R}$ in Eq. (8) results in a matrix $\mathbf{W}_{\mathrm{MLE}}$ that given $\sigma^2_{\mathrm{MLE}}$ attains maximum marginal likelihood. The linear VAE model defined in Eq. (9) is able to recover the global optimum of pPCA when $\mathbf{R} = \mathbf{I}$. Recall from Eq. (6) that $p(\mathbf{z} \mid \mathbf{x})$ is defined in terms of $\mathbf{M} = \mathbf{W}^\top \mathbf{W} + \sigma^2 \mathbf{I}$. When $\mathbf{R} = \mathbf{I}$, we obtain $\mathbf{M} = \mathbf{W}_{\mathrm{MLE}}^\top \mathbf{W}_{\mathrm{MLE}} + \sigma^2_{\mathrm{MLE}} \mathbf{I} = \boldsymbol{\Lambda}_k$, which is diagonal. Thus, setting $\mathbf{V} = \mathbf{M}^{-1} \mathbf{W}_{\mathrm{MLE}}^\top$ and $\mathbf{D} = \sigma^2_{\mathrm{MLE}} \mathbf{M}^{-1} = \sigma^2_{\mathrm{MLE}} \boldsymbol{\Lambda}_k^{-1}$, recovers the true posterior with diagonal covariance at the global optimum. In this case, the ELBO equals the log marginal likelihood and is maximized when the decoder has weights $\mathbf{W} = \mathbf{W}_{\mathrm{MLE}}$. Because the ELBO lower bounds log-likelihood, the global maximum of the ELBO for the linear VAE is the same as the global maximum of the marginal likelihood for pPCA. $\square$

The result of Lemma 1 is somewhat expected because the posterior of pPCA is Gaussian. Further details are given in Appendix C. In addition, we prove a more surprising result that suggests restricting the variational distribution to a Gaussian with a diagonal covariance structure allows one to *identify* the principal components at the global optimum of ELBO.

**Corollary 1.** *The global maximum of the ELBO objective (Eq. (4)) for the linear VAE (Eq. (9)) has the scaled principal components as the columns of the decoder network.*

*Proof.* Follows directly from the proof of Lemma 1 and Eq. (8). $\square$

We discuss this result in Appendix B. This full identifiability is non-trivial and is not achieved even with the regularized linear autoencoder [26].

So far, we have shown that at its global optimum the linear VAE recovers the pPCA solution, which enforces orthogonality of the decoder weight columns. However, the VAE is trained with the ELBO rather than the log marginal likelihood — often using SGD. The majority of existing work suggests that the KL term in the ELBO objective is responsible for posterior collapse. So, we should ask whether this term introduces additional spurious local maxima. Surprisingly, for the linear VAE model the ELBO objective *does not* introduce any additional spurious local maxima. We provide a sketch of the proof below with full details in Appendix C.

**Theorem 1.** *The ELBO objective for a linear VAE does not introduce any additional local maxima to the pPCA model.*

*Proof.* (Sketch) If the decoder has orthogonal columns, then the variational distribution recovers the true posterior at stationary points. Thus, the variational objective will exactly recover the log marginal likelihood. If the decoder does not have orthogonal columns then the variational distribution is no longer tight. However, the ELBO can always be increased by applying an infinitesimal rotation to the right-singular vectors of the decoder towards identity: $\mathbf{W}' \leftarrow \mathbf{W}\mathbf{R}_\epsilon$ (so that the decoder columns are closer to orthogonal). This works because the variational distribution can fit the posterior more closely while the log marginal likelihood is invariant to rotations of the weight columns. Thus, any additional stationary points in the ELBO objective must necessarily be saddle points. $\square$

The theoretical results presented in this section provide new intuition for posterior collapse in VAEs. In particular, the KL between the variational distribution and the prior is not entirely responsible for posterior collapse — log marginal likelihood has a role. The evidence for this is two-fold. We have shown that log marginal likelihood may have spurious local maxima but also that in the linear case the ELBO objective does not add any additional spurious local maxima. Rephrased, in the linear setting the problem lies entirely with the probabilistic model. We should then ask, to what extent do these results hold in the non-linear setting?

# 5 Deep Gaussian VAEs

The deep Gaussian VAE consists of a decoder $D_\theta$ and an encoder $E_\phi$. The ELBO objective can be expressed as,

$$\mathcal{L}(\mathbf{x}; \theta, \phi) = -\mathrm{KL}(q_\phi(\mathbf{z} \mid \mathbf{x}) \parallel p(\mathbf{z})) - \frac{1}{2\sigma^2}\mathbb{E}_{q_\phi(\mathbf{z}|\mathbf{x})}\left[\|D_\theta(\mathbf{z}) - \mathbf{x}\|^2\right] - \frac{1}{2}\log(2\pi\sigma^2) \quad (10)$$

The role of $\sigma^2$ in this objective invites a natural comparison to the $\beta$-VAE objective [18], where the KL term is weighted by $\beta \in \mathbb{R}^+$. Alemi et al. [2] propose using small $\beta$ values to force powerful decoders to utilize the latent variables, but this comes at the cost of poor ELBO. Practitioners must then use downstream task performance for model selection, thus sacrificing one of the primary benefits of likelihood-based models. However, for a given $\beta$, one can find a corresponding $\sigma^2$ (and a learning rate) such that the gradient updates to the network parameters are identical. Importantly, the Gaussian partition function for a Gaussian observation model (the last term on the RHS of Eq. (10)) prevents ELBO from deviating from the $\beta$-VAE's objective with a $\beta$-weighted KL term while maintaining the benefits to representation learning when $\sigma^2$ is small. For the Gaussian VAE, this helps connect the dots between the role of local maxima and observation noise in posterior collapse *vs.* heuristic approaches that attempted to alleviate posterior collapse by diminishing the effect of the KL term [7, 33, 38, 20]. In the following section, we will study the nonlinear VAE empirically and explore connections to the linear theory.

# 6 Experiments

In this section, we present empirical evidence found from studying two distinct claims. First, we verify our theoretical analysis of the linear VAE model. Second, we explore to what extent these insights apply to deep nonlinear VAEs.

## 6.1 Linear VAEs

We ran two sets of experiments on 1000 randomly chosen MNIST images. First, we trained linear VAEs with learnable $\sigma^2$ for a range of hidden dimensions[2]. For each model, we compared the final ELBO to the maximum-likelihood of pPCA finding them to be essentially indistinguishable (as predicted by Lemma 1 and Theorem 1). For the second set of experiments, we took the pPCA MLE solution for $\mathbf{W}$ for each number of hidden dimensions and computed the likelihood under the observation noise which maximizes likelihood for 50 hidden dimensions. We observed that adding additional principal components (after 50) will initially improve likelihood but eventually adding more components (after 200) actually decreases the likelihood. In other words, the collapsed solution is actually preferred if the observation noise is not set correctly — we observe this theoretically through the stability of the stationary points (e.g. Figure 1).

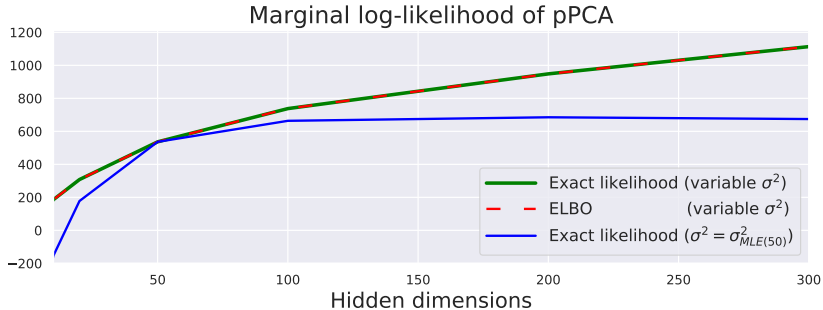

Figure 2: The log marginal likelihood and optimal ELBO of MNIST pPCA solutions over increasing hidden dimension. Green represents the MLE solution (global maximum), the red dashed line is the optimal ELBO solution which matches the global optimum. The blue line shows the log marginal likelihood of the solutions using the full decoder weights when $\sigma^2$ is fixed to its MLE solution for 50 hidden dimensions.

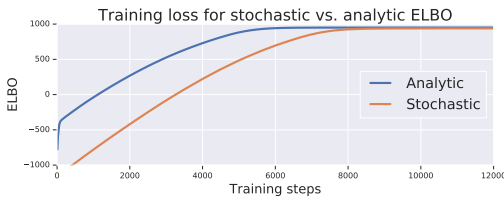
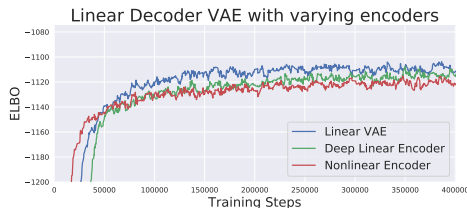

Figure 3: Stochastic vs analytic ELBO training: using the analytic gradient of the ELBO led to faster convergence and better final ELBO (950.7 vs. 939.3).

Figure 4: VAEs with linear decoders trained on real-valued MNIST with nonlinear preprocessing [31]. Final average ELBO on training set are (ordered by legend): -1098.2, -1108.7, -1112.1, -1119.6.

**Effect of stochastic ELBO estimates**    In general, we are unable to compute the ELBO in closed form and so instead rely on unbiased Monte Carlo estimates using the reparameterization trick. These estimates add high-variance noise and can make optimization more challenging [24]. In the linear model, we can compare the solutions obtained using the stochastic ELBO gradients versus the analytic ELBO[3] (Figure 3). Additional experimental details are in Appendix E. We found that stochastic optimization had slower convergence (when compared to analytic training with the same learning rate) and, unsurprisingly, reached a worse final training ELBO value (in other words, worse steady-state risk due to the gradient variance).

**Nonlinear Encoders**    With a linear decoder and nonlinear encoder, Lemma 1 still holds, and the optimal variational distribution is the same as the true posterior has not changed. However, Corollary 1 and Theorem 1 no longer hold in general. Even a deep linear encoder will not have a unique global maximum and new stationary points (possibly maxima) may be introduced to ELBO in general. To investigate how deeper networks may impact optimization of the probabilistic model, we trained linear decoders with varying encoders using ELBO. We do not expect the linear encoder to be outperformed and indeed the empirical results support this (Figure 4).

## 6.2    Investigating posterior collapse in deep nonlinear VAEs

We explored how the analysis of the linear VAEs extends to deep nonlinear models. To do so, we trained VAEs with Gaussian observation models on the MNIST [27] and CelebA [28] datasets. We apply uniform dequantization as in Papamakarios et al. [31] in each case. We also adopt the nonlinear logit preprocessing transformation from Papamakarios et al. [31] to provide fair comparisons with existing work. We also report results of models trained directly in pixel space in the appendix (there is no significant difference for the hypotheses we test).

**Measuring posterior collapse**    In order to measure the extent of posterior collapse, we introduce the following definition. We say that latent dimension dimension $i$ has $(\epsilon, \delta)$-collapsed if $\mathbb{P}_{\mathbf{x} \sim p}[KL(q(z_i|\mathbf{x})||p(z_i)) < \epsilon] \geq 1 - \delta$. Note that the linear VAE can suffer $(0,0)$-collapse. To estimate this practically, we compute the proportion of data samples which induce a variational distribution with KL divergence less than $\epsilon$ and finally report the percentage of dimensions which have $(\epsilon, \delta)$-collapsed. Throughout this work, we fix $\delta = 0.01$ and vary $\epsilon$.

**Investigating $\sigma^2$**    We trained MNIST VAEs with 2 hidden layers in both the decoder and encoder, ReLU activations, and 200 latent dimensions. We first evaluated training with fixed values of the observation noise, $\sigma^2$. This mirrors many public VAE implementations where $\sigma^2$ is fixed to 1 throughout training (also observed by Dai and Wipf [13]), however, our linear analysis suggests that this is suboptimal. Then, we consider the setting where the observation noise and VAE weights are learned simultaneously.

In Table 1 we report the final ELBO of nonlinear VAEs trained on real-valued MNIST. For fixed $\sigma^2$, we found that the final models could have significant differences in ELBO which were maintained even after tuning $\sigma^2$ to the learned representations — the converged representations are less good when $\sigma^2$ is too large as predicted by the linear model. Additionally, we report the final ELBO

| | Model | | ELBO | $\sigma^2$-tuned ELBO | Tuned $\sigma^2$ | Posterior collapse (%) | KL Divergence |
|---|---|---|---|---|---|---|---|
| | Init $\sigma^2$ | Final $\sigma^2$ | | | | | |
| **MNIST** | 10.0 | | $-1450.3 \pm 4.2$ | $-1098.2 \pm 28.3$ | 1.797 | 89.88 | $28.8 \pm 1.4$ |
| | 1.0 | | $-1022.1 \pm 5.4$ | $-1018.3 \pm 5.3$ | 1.145 | 27.38 | $125.4 \pm 4.2$ |
| | 0.1 | | $-3697.3 \pm 493.3$ | $-1190.8 \pm 37.4$ | 0.968 | 3.25 | $368.7 \pm 94.6$ |
| | 0.01 | | $-38612.5 \pm 1189.8$ | $-2090.8 \pm 975.1$ | 0.877 | 0.00 | $695.9 \pm 118.1$ |
| | 0.001 | | $-504259.1 \pm 49149.8$ | $-1744.7 \pm 48.4$ | 0.810 | 0.00 | $756.2 \pm 12.6$ |
| | 10.0 | 1.320 | $-1022.2 \pm 4.5$ | $-1022.3 \pm 4.6$ | 1.318 | 73.75 | $73.8 \pm 9.8$ |
| | 1.0 | 1.183 | $-1011.1 \pm 2.7$ | $-1011.1 \pm 2.8$ | 1.182 | 47.88 | $106.3 \pm 2.5$ |
| | 0.1 | 1.194 | $-1025.4 \pm 8.6$ | $-1025.4 \pm 8.6$ | 1.195 | 29.25 | $116.1 \pm 11.4$ |
| | 0.01 | 1.194 | $-1030.6 \pm 3.5$ | $-1030.5 \pm 3.5$ | 1.191 | 23.00 | $121.9 \pm 7.7$ |
| | 0.001 | 1.208 | $-1038.7 \pm 5.6$ | $-1038.8 \pm 5.6$ | 1.209 | 27.00 | $124.9 \pm 1.6$ |
| **CELEBA 64** | 10.0 | | $-73328.4 \pm 0.49$ | $-55186.7 \pm 35.1$ | 0.2040 | 80.56 | $56.12 \pm 0.4$ |
| | 1.0 | | $-59841.8 \pm 30.1$ | $-51294.8 \pm 333.7$ | 0.1020 | 2.52 | $213.4 \pm 6.3$ |
| | 0.1 | | $-50760.3 \pm 353.4$ | $-50698.5 \pm 393.9$ | 0.0883 | 32.72 | $483.8 \pm 36.2$ |
| | 0.01 | | $-82478.7 \pm 1823.3$ | $-51373.9 \pm 213.3$ | 0.0817 | 0.00 | $1624.2 \pm 8.8$ |
| | 0.001 | | $-531924.5 \pm 17177.6$ | $-57381.5 \pm 512.6$ | 0.0296 | 0.00 | $2680.2 \pm 41.5$ |
| | 10.0 | 0.0962 | $-51109.5 \pm 408.2$ | $-51109.5 \pm 408.3$ | 0.0963 | 53.32 | $364.5 \pm 26.4$ |
| | 1.0 | 0.0875 | $-50631.2 \pm 163.4$ | $-50631.0 \pm 163.3$ | 0.0875 | 54.76 | $462.2 \pm 20.0$ |
| | 0.1 | 0.0863 | $-50646.9 \pm 269.0$ | $-50645.9 \pm 267.5$ | 0.0869 | 28.84 | $520.9 \pm 11.7$ |
| | 0.01 | 0.0911 | $-51285.0 \pm 708.1$ | $-51284.8 \pm 708.1$ | 0.0963 | 5.64 | $557.0 \pm 50.5$ |
| | 0.001 | 0.1040 | $-51695.1 \pm 322.4$ | $-51694.8 \pm 322.7$ | 0.0974 | 0.00 | $537.5 \pm 46.2$ |

Table 1: Evaluation of deep Gaussian VAEs (averaged over 5 trials) on real-valued MNIST. We report the ELBO on the training set in all cases. Collapse percent gives the percentage of latent dimensions which are within 0.01 KL of the prior for at least 99% of the encoder inputs.

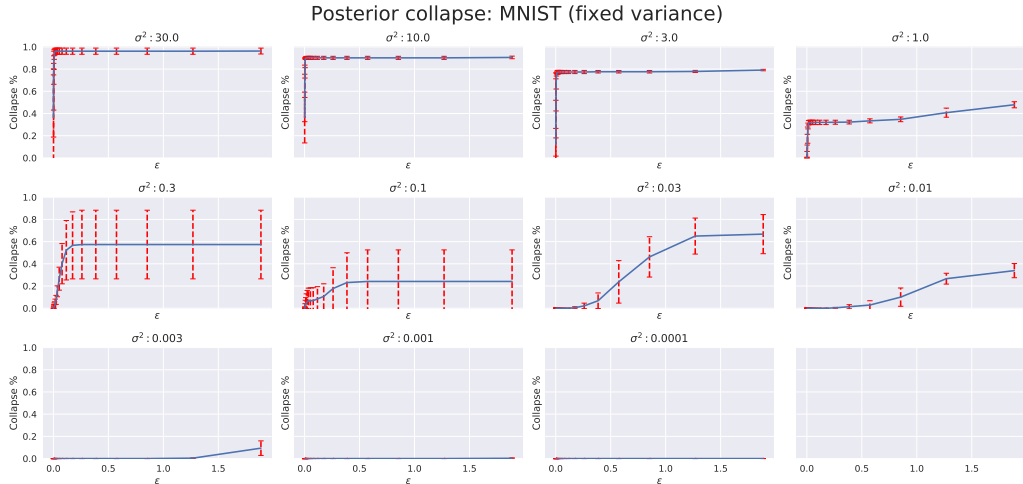

Figure 5: Posterior collapse percentage as a function of $\epsilon$-threshold for a deep VAE trained on MNIST. We measure posterior collapse for trained networks as the proportion of latent dimensions that are within $\epsilon$ KL divergence of the prior for at least a $1 - \delta$ proportion of the training data points ($\delta = 0.01$ in the plots).

values when the model is trained while learning $\sigma^2$ with different initial values of $\sigma^2$. The gap in performance across different initializations is smaller than for fixed $\sigma^2$ but is still significant. The linear VAE does not predict this gap which suggests that learning $\sigma^2$ correctly is more challenging in the nonlinear case.

Despite the large volume of work studying posterior collapse it has not been measured in a consistent way (or even defined so). In Figure 5 and Figure 6 we measure posterior collapse for trained networks as described above (we chose $\delta = 0.01$). By considering a range of $\epsilon$ values we found this was (moderately) robust to stochasticity in data preprocessing. We observed that for large choices of $\sigma^2$ initialization the variational distribution matches the prior closely. This was true even when $\sigma^2$ is learned — suggesting that local optima may contribute to posterior collapse in deep VAEs.

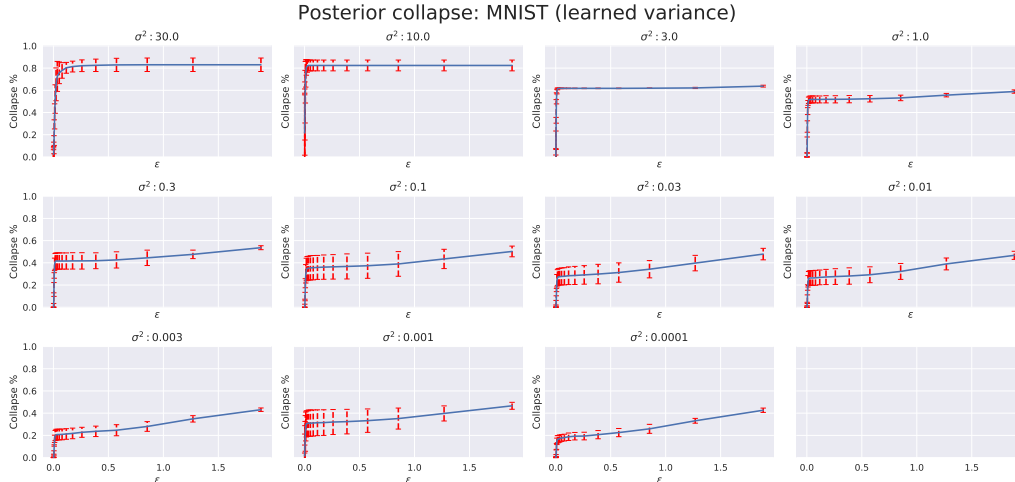

Figure 6: Posterior collapse percentage as a function of $\epsilon$-threshold for a deep VAE trained on MNIST. We measure posterior collapse for trained networks as the proportion of latent dimensions that are within $\epsilon$ KL divergence of the prior for at least a $1 - \delta$ proportion of the training data points ($\delta = 0.01$ in the plots).

**CelebA VAEs**    We trained deep convolutional VAEs with 500 hidden dimensions on images from the CelebA dataset (resized to 64x64). We trained the CelebA VAEs with different fixed values of $\sigma^2$ and compared the ELBO before and after tuning $\sigma^2$ to the learned representations (Table 1). Further, we explored training the CelebA VAE while learning $\sigma^2$ over varied initializations of the observation noise. The VAE is sensitive to the initialization of the observation noise even when $\sigma^2$ is learned (in particular, in terms of the number of collapsed dimensions).

## 7    Discussion

By analyzing the correspondence between linear VAEs and pPCA, this paper makes significant progress towards understanding the causes of posterior collapse. We show that for simple linear VAEs posterior collapse is caused by ill-conditioning of the stationary points in the log marginal likelihood objective. We demonstrate empirically that the same optimization issues play a role in deep non-linear VAEs. Finally, we find that linear VAEs are useful theoretical test-cases for evaluating existing hypotheses on VAEs and we encourage researchers to consider studying their hypotheses in the linear VAE setting.

## 8    Acknowledgements

This work was guided by many conversations with and feedback from our colleagues. In particular, we thank Durk Kingma, Alex Alemi, and Guodong Zhang for invaluable feedback on early versions of this work. RG acknowledges support from the CIFAR Canadian AI Chairs program.

## Footnotes

[2]The VAEs were trained using the analytic ELBO (Appendix C.1) and without mini-batching gradients.

[3]We use 1000 MNIST images, as before, to enable full-batch training so that the only source of noise is from the reparameterization trick [24]

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
