[Supplementary Material]

# A Stationary points of pPCA

Here we briefly summarize the analysis of [39] with some simple additional observations. We recommend that interested readers study Appendix A of Tipping and Bishop [39] for the full details. We begin by formulating the conditions for stationary points of $\sum_{\mathbf{x}_i} \log p(\mathbf{x}_i)$:

$$\mathbf{S}\mathbf{C}^{-1}\mathbf{W} = \mathbf{W} \tag{11}$$

Where $\mathbf{S}$ denotes the sample covariance matrix (assuming we set $\boldsymbol{\mu} = \boldsymbol{\mu}_{MLE}$, *which we do throughout*), and $\mathbf{C} = \mathbf{W}\mathbf{W}^T + \sigma^2 I$ (note that the dimensionality is different to $\mathbf{M}$). There are three possible solutions to this equation, (1) $\mathbf{W} = \mathbf{0}$, (2) $\mathbf{C} = \mathbf{S}$, or (3) the more general solutions. (1) and (2) are not particularly interesting to us, so we focus herein on (3).

We can write $\mathbf{W} = \mathbf{U}\mathbf{L}\mathbf{V}^T$ using its singular value decomposition. Substituting back into the stationary points equation, we recover the following:

$$\mathbf{S}\mathbf{U}\mathbf{L} = \mathbf{U}(\sigma^2 I + \mathbf{L}^2)\mathbf{L} \tag{12}$$

Noting that $\mathbf{L}$ is diagonal, if the $j^{th}$ singular value ($l_j$) is non-zero, this gives $\mathbf{S}\mathbf{u}_j = (\sigma^2 + l_j^2)\mathbf{u}_j$, where $u_j$ is the $j^{th}$ column of $\mathbf{U}$. Thus, $\mathbf{u}_j$ is an eigenvector of $\mathbf{S}$ with eigenvalue $\lambda_j = \sigma^2 + l_j^2$. For $l_j = 0$, $\mathbf{u}_j$ is arbitrary.

Thus, all potential solutions can be written as, $\mathbf{W} = U_q(K_q - \sigma^2 I)^{1/2}\mathbf{R}$, with singular values written as $k_j = \sigma^2$ or $\sigma^2 + l_j^2$ and with $\mathbf{R}$ representing an arbitrary orthogonal matrix.

From this formulation, one can show that the global optimum is attained with $\sigma^2 = \sigma^2_{MLE}$ and $U_q$ and $K_q$ chosen to match the leading singular vectors and values of $\mathbf{S}$.

## A.1 Stability of stationary point solutions

Consider stationary points of the form, $\mathbf{W} = \mathbf{U}_q(K_q - \sigma^2 I)^{1/2}$ where $\mathbf{U}_q$ contains arbitrary eigenvectors of $\mathbf{S}$. In the original pPCA paper they show that all solutions except the leading principal components correspond to saddle points in the optimization landscape. However, this analysis depends critically on $\sigma^2$ being set to the true maximum likelihood estimate. Here we repeat their analysis, considering other (fixed) values of $\sigma^2$.

We consider a small perturbation to a column of $\mathbf{W}$, of the form $\epsilon\mathbf{u}_j$. To analyze the stability of the perturbed solution, we check the sign of the dot-product of the perturbation with the likelihood gradient at $\mathbf{w}_i + \epsilon\mathbf{u}_j$. Ignoring terms in $\epsilon^2$ we can write the dot-product as,

$$\epsilon N(\lambda_j/k_i - 1)\mathbf{u}_j^T\mathbf{C}^{-1}\mathbf{u}_j \tag{13}$$

Now, $\mathbf{C}^{-1}$ is positive definite and so the sign depends only on $\lambda_j/k_i - 1$. The stationary point is stable (local maxima) only if the sign is negative. If $k_i = \lambda_i$ then the maxima is stable only when $\lambda_i > \lambda_j$, in words, the top $q$ principal components are stable. However, we must also consider the case $k = \sigma^2$. Tipping and Bishop [39] show that if $\sigma^2 = \sigma^2_{MLE}$, then this also corresponds to a saddle point as $\sigma^2$ is the average of the smallest eigenvalues meaning some perturbation will be unstable (except in a special case which is handled separately).

However, what happens if $\sigma^2$ is not set to be the maximum likelihood estimate? In this case, it is possible that there are no unstable perturbation directions (that is, $\lambda_j < \sigma^2$ for too many $j$). In this case when $\sigma^2$ is fixed, there are local optima where $\mathbf{W}$ has zero-columns — the same solutions that we observe in non-linear VAEs corresponding to posterior collapse. Note that when $\sigma^2$ is learned in non-degenerate cases the local maxima presented above become saddle points where $\sigma^2$ is made smaller by its gradient. In practice, we find that even when $\sigma^2$ is learned in the non-linear case local maxima exist.

# B Identifiability of the linear VAE

Linear autoencoders suffer from a lack of identifiability which causes the decoder columns to span the principal component subspace instead of recovering it. Kunin et al. [26] showed that adding regularization to the linear autoencoder improves the identifiability — forcing the columns to be identified up to an arbitrary orthogonal transformation, as in pPCA. Here we show that linear VAEs are able to fully identify the principal components.

We once again consider the linear VAE from Eq. (9):

$$p(\mathbf{x} \mid \mathbf{z}) = \mathcal{N}(\mathbf{Wz} + \boldsymbol{\mu}, \sigma^2 \mathbf{I}),$$
$$q(\mathbf{z} \mid \mathbf{x}) = \mathcal{N}(\mathbf{V}(\mathbf{x} - \boldsymbol{\mu}), \mathbf{D}),$$

The output of the VAE, $\tilde{\mathbf{x}}$ is distributed as,

$$\tilde{\mathbf{x}}|\mathbf{x} \sim \mathcal{N}(\mathbf{WV}(\mathbf{x} - \boldsymbol{\mu}) + \boldsymbol{\mu}, \mathbf{WDW}^T).$$

Therefore, the output of the linear VAE is invariant to the following transformation:

$$\mathbf{W} \leftarrow \mathbf{WA},$$
$$\mathbf{V} \leftarrow \mathbf{A}^{-1}\mathbf{V}, \tag{14}$$
$$\mathbf{D} \leftarrow \mathbf{A}^{-1}\mathbf{DA}^{-1},$$

where $\mathbf{A}$ is a diagonal matrix with non-zero entries so that $\mathbf{D}$ is well-defined. However, this transformation changes the variational distribution which affects the loss through the KL term. As argued in Corollary 1, this means that the global optimum is unique for ELBO up to ordering of the eigenvalues/eigenvectors.

At the global optimum, the ordering can be recovered by computing the squared Euclidean norm of the columns of $\mathbf{W}$ (which correspond to the singular values) and ordering according to these quantities.

# C Stationary points of ELBO

Here we present details on the analysis of the stationary points of the ELBO objective. To begin, we first derive closed-form solutions to the components of the log marginal likelihood (including the ELBO). The VAE we focus on is the one presented in Eq. (9), with a linear encoder, linear decoder, Gaussian prior, and Gaussian observation model.

## C.1 Analytic ELBO of the Linear VAE

Remember that one can express the log marginal likelihood as:

$$\log p(\mathbf{x}) = \overset{(A)}{KL(q(\mathbf{z}|\mathbf{x})||p(\mathbf{z}|\mathbf{x}))} - \overset{(B)}{KL(q(\mathbf{z}|\mathbf{x})||p(\mathbf{z}))} + \overset{(C)}{\mathbb{E}_{q(\mathbf{z}|\mathbf{x})}[\log p(\mathbf{x}|\mathbf{z})]}. \tag{15}$$

Each of the terms (A-C) can be expressed in closed form for the linear VAE. Note that the KL term (A) is minimized when the variational distribution is exactly the true posterior distribution. This is possible when the columns of the decoder are orthogonal.

The term (B) can be expressed as,

$$KL(q(\mathbf{z}|\mathbf{x})||p(z)) = 0.5(-\log \det \mathbf{D} + (\mathbf{x} - \boldsymbol{\mu})^T \mathbf{V}^T \mathbf{V}(\mathbf{x} - \boldsymbol{\mu}) + tr(\mathbf{D}) - q). \tag{16}$$

The term (C) can be expressed as,

$$\mathbb{E}_{q(\mathbf{z}|\mathbf{x})}[\log p(\mathbf{x}|\mathbf{z})] = \mathbb{E}_{q(\mathbf{z}|\mathbf{x})}\left[-(\mathbf{Wz} - (\mathbf{x} - \boldsymbol{\mu}))^T(\mathbf{Wz} - (\mathbf{x} - \boldsymbol{\mu}))/2\sigma^2 - \frac{d}{2}\log 2\pi\sigma^2\right] \tag{17}$$

$$= \mathbb{E}_{q(\mathbf{z}|\mathbf{x})}\left[\frac{-(\mathbf{Wz})^T(\mathbf{Wz}) + 2(\mathbf{x} - \boldsymbol{\mu})^T \mathbf{Wz} - (\mathbf{x} - \boldsymbol{\mu})^T(\mathbf{x} - \boldsymbol{\mu})}{2\sigma^2} - \frac{d}{2}\log 2\pi\sigma^2\right]. \tag{18}$$

Noting that $\mathbf{Wz} \sim \mathcal{N}\left(\mathbf{WV}(\mathbf{x} - \boldsymbol{\mu}), \mathbf{WDW}^T\right)$, we can compute the expectation analytically and obtain,

$$\mathbb{E}_{q(\mathbf{z}|\mathbf{x})}\left[\log p(\mathbf{x}|\mathbf{z})\right] = \frac{1}{2\sigma^2}[-tr(\mathbf{WDW}^T) - (\mathbf{x} - \boldsymbol{\mu})^T\mathbf{V}^T\mathbf{W}^T\mathbf{WV}(\mathbf{x} - \boldsymbol{\mu}) \tag{19}$$

$$+ 2(\mathbf{x} - \boldsymbol{\mu})^T\mathbf{WV}(\mathbf{x} - \boldsymbol{\mu}) - (\mathbf{x} - \boldsymbol{\mu})^T(\mathbf{x} - \boldsymbol{\mu})] - \frac{d}{2}\log 2\pi\sigma^2. \tag{20}$$

## C.2 Finding stationary points

To compute the stationary points we must take derivatives with respect to $\boldsymbol{\mu}, \mathbf{D}, \mathbf{W}, \mathbf{V}, \sigma^2$. As before, we have $\boldsymbol{\mu} = \boldsymbol{\mu}_{MLE}$ at the global maximum and for simplicity we fix $\boldsymbol{\mu}$ here for the remainder of the analysis.

Taking the marginal likelihood over the whole dataset, at the stationary points we have,

$$\frac{\partial}{\partial \mathbf{D}}(-(B) + (C)) = \frac{N}{2}(\mathbf{D}^{-1} - \mathbf{I} - \frac{1}{\sigma^2}\text{diag}(\mathbf{W}^T\mathbf{W})) = 0 \tag{21}$$

$$\frac{\partial}{\partial \mathbf{V}}(-(B) + (C)) = \frac{N}{\sigma^2}(\mathbf{W}^T - (\mathbf{W}^T\mathbf{W} + \sigma^2\mathbf{I})\mathbf{V})\mathbf{S} = 0 \tag{22}$$

$$\frac{\partial}{\partial \mathbf{W}}(-(B) + (C)) = \frac{N}{\sigma^2}(\mathbf{SV}^T - \mathbf{DW} - \mathbf{WVSV}^T) = 0 \tag{23}$$

The above are computed using standard matrix derivative identities [32]. These equations yield the expected solution for the variational distribution directly. From Eq. (21) we compute $\mathbf{D}^* = \sigma^2(\text{diag}(\mathbf{W}^T\mathbf{W}) + \sigma^2\mathbf{I})^{-1}$ and $\mathbf{V}^* = \mathbf{M}^{-1}\mathbf{W}^T$, recovering the true posterior mean in all cases and getting the correct posterior covariance when the columns of $\mathbf{W}$ are orthogonal. We will now proceed with the proof of Theorem 1.

**Theorem 1.** *The ELBO objective for a linear VAE does not introduce any additional local maxima to the pPCA model.*

*Proof.* If the columns of $\mathbf{W}$ are orthogonal then the log marginal likelihood is recovered exactly at all stationary points. This is a direct consequence of the posterior mean *and* covariance being recovered exactly at all stationary points so that (1) is zero.

We must give separate treatment to the case where there is a stationary point without orthogonal columns of $\mathbf{W}$. Suppose we have such a stationary point, using the singular value decomposition we can write $\mathbf{W} = \mathbf{ULR}^T$, where $\mathbf{U}$ and $\mathbf{R}$ are orthogonal matrices. Note that $\log p(\mathbf{x})$ is invariant to the choice of $\mathbf{R}$ [39]. However, the choice of $\mathbf{R}$ does affect the first term (1) of Eq. (15): this term is minimized when $\mathbf{R} = \mathbf{I}$, and thus the ELBO must increase.

To formalize this argument, we compute (1) at a stationary point. From above, at every stationary point the mean of the variational distribution exactly matches the true posterior. Thus the KL simplifies to:

$$KL(q(\mathbf{z}|\mathbf{x})||p(\mathbf{z}|\mathbf{x})) = \frac{1}{2}\left(tr(\frac{1}{\sigma^2}\mathbf{MD}) - q + q\log\sigma^2 - \log(\det\mathbf{M}\det\mathbf{D})\right), \tag{24}$$

$$= \frac{1}{2}\left(tr(\mathbf{M}\widetilde{\mathbf{M}}^{-1}) - q - \log\frac{\det\mathbf{M}}{\det\widetilde{\mathbf{M}}}\right), \tag{25}$$

$$= \frac{1}{2}\left(\sum_{i=1}^{q}\frac{\mathbf{M}_{ii}}{\widetilde{\mathbf{M}}_{ii}} - q - \log\det\mathbf{M} + \log\det\widetilde{\mathbf{M}}\right), \tag{26}$$

$$= \frac{1}{2}\left(\log\det\widetilde{\mathbf{M}} - \log\det\mathbf{M}\right), \tag{27}$$

$$\tag{28}$$

where $\widetilde{\mathbf{M}} = \text{diag}(\mathbf{W}^T\mathbf{W}) + \sigma^2\mathbf{I}$. Now consider applying a small rotation to $\mathbf{W}$: $\mathbf{W} \mapsto \mathbf{WR}_\epsilon$. As the optimal $\mathbf{D}$ and $\mathbf{V}$ are continuous functions of $\mathbf{W}$, this corresponds to a small perturbation of

these parameters too for a sufficiently small rotation. Importantly, $\log \det \mathbf{M}$ remains fixed for any orthogonal choice of $\mathbf{R}_\epsilon$ but $\log \det \widetilde{\mathbf{M}}$ does not. Thus, we choose $\mathbf{R}_\epsilon$ to minimize this term. In this manner, (1) shrinks meaning that the ELBO (-2)+(3) must increase. Thus if the stationary point existed, it must have been a saddle point.

We now describe how to construct such a small rotation matrix. First note that without loss of generality we can assume that $\det(\mathbf{R}) = 1$. (Otherwise, we can flip the sign of a column of $\mathbf{R}$ and the corresponding column of $\mathbf{U}$.) And additionally, we have $\mathbf{WR} = \mathbf{UL}$, which is orthogonal.

The Special Orthogonal group of determinant 1 orthogonal matrices is a compact, connected Lie group and therefore the exponential map from its Lie algebra is surjective. This means that we can find an upper-triangular matrix $\mathbf{B}$, such that $\mathbf{R} = \exp\{\mathbf{B} - \mathbf{B}^T\}$. Consider $\mathbf{R}_\epsilon = \exp\{\frac{1}{n(\epsilon)}(\mathbf{B} - \mathbf{B}^T)\}$, where $n(\epsilon)$ is an integer chosen to ensure that the elements of $\mathbf{B}$ are within $\epsilon > 0$ of zero. This matrix is a rotation in the direction of $\mathbf{R}$ which we can make arbitrarily close to the identity by a suitable choice of $\epsilon$. This is verified through the Taylor series expansion of $\mathbf{R}_\epsilon = I + \frac{1}{n(\epsilon)}(\mathbf{B} - \mathbf{B}^T) + O(\epsilon^2)$. Thus, we have identified a small perturbation to $\mathbf{W}$ (and $\mathbf{D}$ and $\mathbf{V}$) which decreases the posterior KL (A) but keeps the log marginal likelihood constant. Thus, the ELBO increases and the stationary point must be a saddle point.

$\square$

### C.3  Bernoulli Probabilistic PCA

We would like to extend our linear analysis to the case where we have a Bernoulli observation model, as this setting also suffers severely from posterior collapse. The analysis may also shed light on more general categorical observation models which have also been used. Typically, in these settings a continuous latent space is still used (for example, Bowman et al. [7]).

We will consider the following model,

$$
\begin{aligned}
p(\mathbf{z}) &= \mathcal{N}(0, \mathbf{I}), \\
p(\mathbf{x}|\mathbf{z}) &= \text{Bernoulli}(\mathbf{y}), \\
\mathbf{y} &= \sigma(\mathbf{Wz} + \boldsymbol{\mu})
\end{aligned}
\tag{29}
$$

where $\sigma$ denotes the sigmoid function, $\sigma(y) = 1/(1 + \exp(-y))$ and we assume an independent Bernoulli observation model over $\mathbf{x}$.

Unfortunately, under this model it is difficult to reason about the stationary points. There is no closed form solution for the marginal likelihood $p(\mathbf{x})$ or the posterior distribution $p(\mathbf{z}|\mathbf{x})$. Numerical integration methods exist which may make it easy to evaluate this quantity in practice but they will not immediately provide us a good gradient signal.

We can compute the density function for $\mathbf{y}$ using the change of variables formula. Noting that $\mathbf{Wz} + \boldsymbol{\mu} \sim \mathcal{N}(\boldsymbol{\mu}, \mathbf{WW}^T)$, we recover the following logit-Normal distribution:

$$
f(\mathbf{y}) = \frac{1}{\sqrt{2\pi|\mathbf{WW}^T|}} \frac{1}{\Pi_i y_i(1 - y_i)} \exp\{-\frac{1}{2}\left(\log(\frac{\mathbf{y}}{1 - \mathbf{y}}) - \boldsymbol{\mu}\right)^T (\mathbf{WW}^T)^{-1} \left(\log(\frac{\mathbf{y}}{1 - \mathbf{y}}) - \boldsymbol{\mu}\right)\}
\tag{30}
$$

We can write the marginal likelihood as,

$$
p(\mathbf{x}) = \int p(\mathbf{x}|\mathbf{z})p(\mathbf{z})d\mathbf{z},
\tag{31}
$$

$$
= \mathbb{E}_{\mathbf{z}}\left[\mathbf{y}(\mathbf{z})^{\mathbf{x}}(1 - \mathbf{y}(\mathbf{z}))^{1-\mathbf{x}}\right],
\tag{32}
$$

where $(\cdot)^{\mathbf{x}}$ is taken to be elementwise. Unfortunately, the expectation of a logit-normal distribution has no closed form [3] and so we cannot tractably compute the marginal likelihood.

Similarly, under ELBO we need to compute the expected reconstruction error. This can be written as,

$$\mathbb{E}_{q(\mathbf{z}|\mathbf{x})}[\log p(\mathbf{x}|\mathbf{z})] = \int \mathbf{y}(\mathbf{z})^{\mathbf{x}}(1 - \mathbf{y}(\mathbf{z}))^{1-\mathbf{x}}\mathcal{N}(\mathbf{z}; \mathbf{V}(\mathbf{x} - \boldsymbol{\mu}), \mathbf{D})d\mathbf{z}, \qquad (33)$$

another intractable integral.

## D    Related Work (Extended)

Due to the large volume of work studying posterior collapse in variational autoencoders, we have included here an extended discussion of related work. We utilize this additional space to provide a more in-depth discussion of the related work presented in the main paper and to highlight additional work.

Tomczak and Welling [40] introduce the VampPrior, a hierarchical learned prior for VAEs. Tomczak and Welling [40] show empirically that such a learned prior can mitigate posterior collapse (which they refer to as inactive stochastic units). While the authors provide limited theoretical support for the efficacy of their method in reducing posterior collapse, they claim intuitively that by enabling multi-modal prior distributions the KL term is less likely to force inactive units — possibly by reducing the impact of local optima corresponding to posterior collapse.

In the main paper we discuss the work of Dai et al. [14], which connect robust PCA methods and VAEs. In particular, Section 2 of their manuscript studies the case of a linear decoder and shows that, when the encoder takes the form of the optimal variational distribution, the ELBO of the resulting VAE collapses into the pPCA objective. We study the ELBO without optimality assumptions on the linear encoder and characterize the optimization landscape with no additional assumptions. They claim further that all minima of the (encoder-optimal) ELBO objective are globally optimal — we show in fact that for a linear encoder there is a fully identifiable global optimum.

Dai and Wipf [13] discuss the important of the observation noise, and in fact show that under some assumptions the optimal observation noise should shrink to zero (Theorem 4 in their work). These assumptions amount to the number of latent dimensions exceeding the dimensionality of the true data manifold. However, in the linear model (whose latent dimensions do not exceed the input space dimensionality) the optimal variance does not shrink towards zero and is instead given by the sum of the variance lost in the linear projection. Note that this does not violate the results of Dai and Wipf [13], but highlights the need to consider model capacity against data complexity, as in Alemi et al. [2].

## E    Experiment details

We used Tensorflow [1] for our experiments with linear and deep VAEs. In each case, the models were trained using a single GPU.

**Visualizing stationary points of pPCA**    For this experiment we computed the pPCA MLE using a subset of 1000 random training images from the MNIST dataset. We evaluate and plot the log marginal likelihood in closed form on this same subset. In this case, we did not dequantize or apply any nonlinear processing to the data.

**Stochastic vs. Analytic VAE**    We trained linear VAEs with 200 hidden dimensions. We used full-batch training with 1000 MNIST digits samples randomly from the training set (the same data as used to produce Figure 2). We trained each model with the Adam optimizer and a fixed learning rate, grid searching to find the learning rate which gave the best ELBO after 12000 training steps in the range $\{0.0001, 0.0003, 0.001, 0.003\}$. For both models, 0.001 provided the best final ELBO.

**MNIST VAE**    The VAEs we trained on MNIST all had the same architecture: 784-1024-512-k-512-1024-784. The Gaussian likelihood is fairly uncommon for this dataset, which is nearly binary, but it provides a good setting for us to investigate our theoretical findings. To dequantize the data, we added uniform random noise and rescaled the pixel values to be in the range $[0, 1]$. We then applied a nonlinear logistic transform as in [31]. The VAE parameters were optimized jointly using the Adam optimizer [23]. We trained the VAE for 1000 epochs total, keeping the learning rate fixed throughout.

Figure 7: Proportion of inactive units thresholded by KL divergence when using 0-1 KL-annealing and a fixed value of $\sigma^2$. The solid line represents the final model while the dashed line is the model after only 80 epochs of training. KL annealing reduces posterior collapse during the early stages of training but ultimately fails to escape these sub-optimal solutions as the KL weight is increased.

Figure 8: Comparing learned solutions using KL-Annealing versus standard ELBO training when $\sigma^2$ is learned.

We performed a grid search over learning rates in the range $\{0.0001, 0.0003, 0.001, 0.003\}$ and reported results for the model which achieved the best training ELBO.

**CelebA VAE** We used the convolutional architecture proposed by Higgins et al. [18] trained on 64x64 images from the CelebA dataset [28]. Otherwise, the experimental procedure followed that of the MNIST VAEs with the nonlinear preprocessing hyperparameters set as in [31].

## E.1 Additional results

### E.1.1 Evaluating KL Annealing

We found that KL-annealing may provide temporary relief from posterior collapse but that if $\sigma^2$ is not learned simultaneously then the collapsed solution is recovered. In Figure 7 we show the proportion of units collapsed by threshold for several fixed choices of $\sigma^2$ when $\beta$ is annealed from 0 to 1 over the first 100 epochs. The solid lines correspond to the final model while the dashed line corresponds to the model at 80 epochs of training. KL-annealing was able to reduce posterior collapse initially but eventually fell back to the collapsed solution.

After finding that KL-annealing alone was insufficient to prevent posterior collapse we explored KL annealing while learning $\sigma^2$. Based on our analysis in the linear case we expect that this should work well: while $\beta$ is small the model should be able to learn to reduce $\sigma^2$. We trained using the same KL schedule and also with standard ELBO while learning $\sigma^2$. The results are presented in Figure 8 and Figure 9. Under the ELBO objective, $\sigma^2$ is reduced somewhat but ultimately a large degree of posterior collapse is present. Using KL-annealing, the VAE is able to learn a much smaller $\sigma^2$ value and ultimately reduces posterior collapse. This suggests that the non-linear VAE dynamics may be similar to the linear case when suitably conditioned.

Figure 9: Learning $\sigma^2$ for CelebA VAEs with standard ELBO training and KL-Annealing. KL-Annealing enables a smaller $\sigma^2$ to be learned and reduces posterior collapse.

| | Model | | ELBO | $\sigma^2$-tuned ELBO | Tuned $\sigma^2$ | Posterior collapse (%) | KL Divergence |
|---|---|---|---|---|---|---|---|
| | Init $\sigma^2$ | Final $\sigma^2$ | | | | | |
| MNIST | 30.0 | | $-1850.4 \pm 29.0$ | $-1374.9 \pm 199.0$ | 4.451 | 95.0 | $10.9 \pm 6.7$ |
| | 10.0 | | $-1450.3 \pm 4.2$ | $-1098.2 \pm 28.3$ | 1.797 | 89.88 | $28.8 \pm 1.4$ |
| | 3.0 | | $-1114.9 \pm 1.1$ | $-1018.8 \pm 1.0$ | 1.361 | 76.75 | $58.5 \pm 1.4$ |
| | 1.0 | | $-1022.1 \pm 5.4$ | $-1018.3 \pm 5.3$ | 1.145 | 27.38 | $125.4 \pm 4.2$ |
| | 0.3 | | $-1816.7 \pm 270.6$ | $-1104.6 \pm 6.2$ | 1.275 | 2.0 | $179.3 \pm 85.9$ |
| | 0.1 | | $-3697.3 \pm 493.3$ | $-1190.8 \pm 37.4$ | 0.968 | 3.25 | $368.7 \pm 94.6$ |
| | 0.03 | | $-18549.3 \pm 4892.0$ | $-1283.2 \pm 63.3$ | 1.47 | 0.0 | $305.3 \pm 75.4$ |
| | 0.01 | | $-38612.5 \pm 1189.8$ | $-1403.1 \pm 21.0$ | 1.006 | 0.0 | $560.9 \pm 32.4$ |
| | 0.003 | | $-139538.8 \pm 21148.5$ | $-2090.8 \pm 975.1$ | 0.877 | 0.0 | $695.9 \pm 118.1$ |
| | 0.001 | | $-504259.1 \pm 49149.8$ | $-1744.7 \pm 48.4$ | 0.81 | 0.0 | $756.2 \pm 12.6$ |
| | 30.0 | 1.478 | $-1060.9 \pm 23.1$ | $-1061.0 \pm 23.0$ | 1.476 | 33.75 | $70.9 \pm 13.8$ |
| | 10.0 | 1.32 | $-1022.2 \pm 4.5$ | $-1022.3 \pm 4.6$ | 1.318 | 73.75 | $73.8 \pm 9.8$ |
| | 3.0 | 1.178 | $-1004.6 \pm 1.4$ | $-1004.5 \pm 1.3$ | 1.181 | 58.38 | $99.8 \pm 1.5$ |
| | 1.0 | 1.183 | $-1011.1 \pm 2.7$ | $-1011.1 \pm 2.8$ | 1.182 | 47.88 | $106.3 \pm 2.5$ |
| | 0.3 | 1.195 | $-1020.0 \pm 6.0$ | $-1019.9 \pm 6.1$ | 1.191 | 37.75 | $111.6 \pm 6.1$ |
| | 0.1 | 1.194 | $-1025.4 \pm 8.6$ | $-1025.4 \pm 8.6$ | 1.195 | 29.25 | $116.1 \pm 11.4$ |
| | 0.03 | 1.197 | $-1030.6 \pm 6.6$ | $-1030.5 \pm 6.6$ | 1.198 | 22.62 | $120.2 \pm 10.5$ |
| | 0.01 | 1.194 | $-1030.6 \pm 3.5$ | $-1030.5 \pm 3.5$ | 1.191 | 23.0 | $121.9 \pm 7.7$ |
| | 0.003 | 1.19 | $-1033.7 \pm 2.3$ | $-1033.6 \pm 2.3$ | 1.187 | 16.62 | $126.4 \pm 6.8$ |
| | 0.001 | 1.208 | $-1038.7 \pm 5.6$ | $-1038.8 \pm 5.6$ | 1.209 | 27.0 | $124.9 \pm 1.6$ |

Table 2: Full evaluation of deep Gaussian VAEs (averaged over 5 trials) on real-valued MNIST with nonlinear preprocessing [31]. Collapse percent gives the percentage of latent dimensions which are within 0.01 KL of the prior for at least 99% of the encoder inputs.

### E.1.2 Full results tables

### E.1.3 Qualitative Results

Reconstructions from the KL-Annealed CelebA model are shown in Figure 12. We also show the output of interpolating in the latent space in Figure 13. To produce the latter plot, we compute the variational mean of 3 input points (top left, top right, bottom left) and interpolate linearly on the plane between them. We also extrapolate out to a fourth point (bottom right), which lies on the plane defined by the other points.

Figure 10: Posterior collapse percentage as a function of $\epsilon$-threshold for a deep VAE trained on CelebA with fixed $\sigma^2$. We measure posterior collapse for trained networks as the proportion of latent dimensions that are within $\epsilon$ KL divergence of the prior for at least a $1 - \delta$ proportion of the training data points ($\delta = 0.01$ in the plots).

Figure 11: Posterior collapse percentage as a function of $\epsilon$-threshold for a deep VAE trained on CelebA with learned $\sigma^2$. We measure posterior collapse for trained networks as the proportion of latent dimensions that are within $\epsilon$ KL divergence of the prior for at least a $1 - \delta$ proportion of the training data points ($\delta = 0.01$ in the plots).

Figure 12: Reconstructions from the convolutional VAE trained with KL-Annealing on CelebA.

| | Model | | ELBO | $\sigma^2$-tuned ELBO | Tuned $\sigma^2$ | Posterior | KL |
|---|---|---|---|---|---|---|---|
| | Init $\sigma^2$ | Final $\sigma^2$ | | | | collapse (%) | Divergence |
| CELEBA 64 | 30.0 | | $-79986.2 \pm 0.10$ | $-57883.8 \pm 19.3$ | 0.423 | 93.68 | $26.0 \pm 0.19$ |
| | 10.0 | | $-73328.4 \pm 0.49$ | $-55186.7 \pm 35.1$ | 0.204 | 80.56 | $56.12 \pm 0.42$ |
| | 3.0 | | $-66145.6 \pm 2.44$ | $-52828.5 \pm 58.6$ | 0.132 | 20.64 | $120.4 \pm 1.37$ |
| | 1.0 | | $-59841.8 \pm 30.1$ | $-51294.8 \pm 333.7$ | 0.102 | 2.52 | $213.4 \pm 6.3$ |
| | 0.3 | | $-54370.4 \pm 849.9$ | $-52155.2 \pm 1855.2$ | 0.122 | 74.52 | $267.2 \pm 51.9$ |
| | 0.1 | | $-50760.3 \pm 353.4$ | $-50698.5 \pm 393.9$ | 0.0883 | 32.72 | $483.8 \pm 36.2$ |
| | 0.03 | | $-64322.8 \pm 312.9$ | $-58077.9 \pm 206.2$ | 0.0463 | 0.0 | $1521.1 \pm 11.6$ |
| | 0.01 | | $-82478.7 \pm 1823.3$ | $-51373.9 \pm 213.3$ | 0.0817 | 0.0 | $1624.2 \pm 8.78$ |
| | 0.003 | | $-192967.7 \pm 4410.4$ | $-51978.4 \pm 159.3$ | 0.0685 | 0.0 | $2108.4 \pm 26.2$ |
| | 0.001 | | $-531924.5 \pm 17177.6$ | $-57381.5 \pm 512.6$ | 0.0296 | 0.0 | $2680.2 \pm 41.45$ |
| | 30.0 | 0.478 | $-57773.0 \pm 3622.9$ | $-56068.5 \pm 2771.0$ | 0.475 | 14.2 | $221.7 \pm 99.0$ |
| | 10.0 | 0.0962 | $-51109.5 \pm 408.2$ | $-51109.5 \pm 408.3$ | 0.0963 | 53.32 | $364.5 \pm 26.4$ |
| | 3.0 | 0.0891 | $-50813.2 \pm 229.7$ | $-50813.3 \pm 229.7$ | 0.0889 | 10.96 | $545.2 \pm 5.5$ |
| | 1.0 | 0.0875 | $-50631.2 \pm 163.4$ | $-50631.0 \pm 163.3$ | 0.0875 | 54.76 | $462.2 \pm 20.0$ |
| | 0.3 | 0.0890 | $-50963.4 \pm 331.2$ | $-50963.2 \pm 331.3$ | 0.0892 | 7.96 | $670.7 \pm 79.2$ |
| | 0.1 | 0.0863 | $-50646.9 \pm 269.0$ | $-50645.9 \pm 267.5$ | 0.0869 | 28.84 | $520.9 \pm 11.7$ |
| | 0.03 | 0.121 | $-53263.4 \pm 71.5$ | $-53263.3 \pm 71.3$ | 0.126 | 0.0 | $856.2 \pm 19.7$ |
| | 0.01 | 0.0911 | $-51285.0 \pm 708.1$ | $-51284.8 \pm 708.1$ | 0.0963 | 5.64 | $557.0 \pm 50.5$ |
| | 0.003 | 0.0952 | $-51056.4 \pm 1216.9$ | $-51055.9 \pm 1217.4$ | 0.094 | 0.8 | $577.4 \pm 30.4$ |
| | 0.001 | 0.104 | $-51695.1 \pm 322.4$ | $-51694.8 \pm 322.7$ | 0.0974 | 0.0 | $537.5 \pm 46.2$ |

Table 3: Full evaluation of deep Gaussian VAEs (averaged over 5 trials) on real-valued CelebA with nonlinear preprocessing [31]. Collapse percent gives the percentage of latent dimensions which are within 0.01 KL of the prior for at least 99% of the encoder inputs.

| | Model | | ELBO | $\sigma^2$-tuned ELBO | Tuned $\sigma^2$ | Posterior | KL |
|---|---|---|---|---|---|---|---|
| | Init $\sigma^2$ | Final $\sigma^2$ | | | | collapse (%) | Divergence |
| MNIST | 30.0 | | $-6402.0 \pm 0.0$ | $-6248.4 \pm 197.2$ | 22.323 | 0.0 | $0.0 \pm 0.0$ |
| | 10.0 | | $-5973.1 \pm 0.0$ | $-5821.0 \pm 194.6$ | 7.443 | 0.0 | $0.0 \pm 0.0$ |
| | 3.0 | | $-5507.1 \pm 0.1$ | $-5360.4 \pm 185.4$ | 2.235 | 1.7 | $0.6 \pm 0.3$ |
| | 1.0 | | $-5087.9 \pm 3.1$ | $-4954.7 \pm 156.9$ | 0.747 | 0.0 | $4.5 \pm 2.3$ |
| | 0.3 | | $-4638.4 \pm 3.6$ | $-4516.8 \pm 137.9$ | 0.225 | 0.0 | $12.5 \pm 1.5$ |
| | 0.1 | | $-4243.1 \pm 17.6$ | $-4154.6 \pm 62.1$ | 0.076 | 0.0 | $25.6 \pm 3.0$ |
| | 0.03 | | $-3820.7 \pm 13.9$ | $-3785.2 \pm 26.6$ | 0.027 | 0.0 | $55.8 \pm 2.1$ |
| | 0.01 | | $-3508.4 \pm 12.3$ | $-3483.5 \pm 13.1$ | 0.009 | 0.0 | $112.8 \pm 6.7$ |
| | 0.003 | | $-3267.3 \pm 2.6$ | $-3247.1 \pm 2.8$ | 0.003 | 0.0 | $252.2 \pm 2.1$ |
| | 0.001 | | $-3137.7 \pm 5.2$ | $-3136.7 \pm 5.4$ | 0.001 | 0.0 | $422.7 \pm 2.6$ |
| | 30.0 | 0.067 | $-4398.7 \pm 0.0$ | $-4398.7 \pm 0.0$ | 0.067 | 0.0 | $0.0 \pm 0.0$ |
| | 10.0 | 0.044 | $-4146.3 \pm 309.2$ | $-4146.3 \pm 309.2$ | 0.044 | 0.0 | $30.1 \pm 36.9$ |
| | 3.0 | 0.01 | $-3736.3 \pm 14.3$ | $-3736.4 \pm 14.3$ | 0.01 | 0.0 | $73.7 \pm 1.9$ |
| | 1.0 | 0.008 | $-3673.0 \pm 17.7$ | $-3672.9 \pm 17.7$ | 0.008 | 0.0 | $85.2 \pm 2.5$ |
| | 0.3 | 0.006 | $-3569.8 \pm 26.4$ | $-3569.8 \pm 26.4$ | 0.006 | 0.0 | $100.8 \pm 3.7$ |
| | 0.1 | 0.003 | $-3355.8 \pm 7.6$ | $-3355.8 \pm 7.6$ | 0.003 | 0.0 | $151.7 \pm 2.4$ |
| | 0.03 | 0.001 | $-3138.9 \pm 10.6$ | $-3139.0 \pm 10.6$ | 0.001 | 0.0 | $275.4 \pm 3.1$ |
| | 0.01 | 0.001 | $-3126.1 \pm 5.0$ | $-3126.1 \pm 5.0$ | 0.001 | 0.0 | $349.3 \pm 5.4$ |
| | 0.003 | 0.001 | $-3161.4 \pm 4.0$ | $-3161.3 \pm 4.0$ | 0.001 | 0.0 | $373.5 \pm 7.5$ |
| | 0.001 | 0.001 | $-3145.4 \pm 6.1$ | $-3145.4 \pm 6.1$ | 0.001 | 0.0 | $378.4 \pm 7.7$ |

Table 4: Evaluation of deep Gaussian VAEs (averaged over 5 trials) on real-valued MNIST without any nonlinear preprocessing. Collapse percent gives the percentage of latent dimensions which are within 0.01 KL of the prior for at least 99% of the encoder inputs.

| Model | | ELBO | $\sigma^2$-tuned ELBO | Tuned $\sigma^2$ | Posterior collapse (%) | KL Divergence |
|---|---|---|---|---|---|---|
| Init $\sigma^2$ | Final $\sigma^2$ | | | | | |
| 30.0 | | $-79986.2 \pm 0.10$ | $-57883.8 \pm 19.3$ | 0.423 | 93.68 | $26.0 \pm 0.19$ |
| 10.0 | | $-73328.4 \pm 0.49$ | $-55186.7 \pm 35.1$ | 0.204 | 80.56 | $56.12 \pm 0.42$ |
| 3.0 | | $-66145.6 \pm 2.44$ | $-52828.5 \pm 58.6$ | 0.132 | 20.64 | $120.4 \pm 1.37$ |
| 1.0 | | $-59841.8 \pm 30.1$ | $-51294.8 \pm 333.7$ | 0.102 | 2.52 | $213.4 \pm 6.3$ |
| 0.3 | | $-54370.4 \pm 849.9$ | $-52155.2 \pm 1855.2$ | 0.122 | 74.52 | $267.2 \pm 51.9$ |
| 0.1 | | $-50760.3 \pm 353.4$ | $-50698.5 \pm 393.9$ | 0.0883 | 32.72 | $483.8 \pm 36.2$ |
| 0.03 | | $-64322.8 \pm 312.9$ | $-58077.9 \pm 206.2$ | 0.0463 | 0.0 | $1521.1 \pm 11.6$ |
| 0.01 | | $-82478.7 \pm 1823.3$ | $-51373.9 \pm 213.3$ | 0.0817 | 0.0 | $1624.2 \pm 8.78$ |
| 0.003 | | $-192967.7 \pm 4410.4$ | $-51978.4 \pm 159.3$ | 0.0685 | 0.0 | $2108.4 \pm 26.2$ |
| 0.001 | | $-531924.5 \pm 17177.6$ | $-57381.5 \pm 512.6$ | 0.0296 | 0.0 | $2680.2 \pm 41.45$ |
| 30.0 | 0.005 | $-53179.6 \pm 450.2$ | $-53179.6 \pm 450.3$ | 0.005 | 0.0 | $302.8 \pm 29.8$ |
| 10.0 | 0.004 | $-51748.5 \pm 178.2$ | $-51748.5 \pm 178.2$ | 0.004 | 0.0 | $482.3 \pm 24.7$ |
| 3.0 | 0.004 | $-51548.9 \pm 154.1$ | $-51548.9 \pm 154.2$ | 0.004 | 0.0 | $489.5 \pm 21.8$ |
| 1.0 | 0.004 | $-51356.9 \pm 79.1$ | $-51356.9 \pm 79.1$ | 0.004 | 0.0 | $516.3 \pm 18.0$ |
| 0.3 | 0.004 | $-51767.7 \pm 369.2$ | $-51767.7 \pm 369.1$ | 0.004 | 22.0 | $439.7 \pm 33.3$ |
| 0.1 | 0.004 | $-51637.3 \pm 163.3$ | $-51637.1 \pm 163.5$ | 0.004 | 0.0 | $577.3 \pm 13.5$ |
| 0.03 | 0.004 | $-51792.6 \pm 163.4$ | $-51792.6 \pm 163.6$ | 0.004 | 45.48 | $484.6 \pm 22.6$ |
| 0.01 | 0.004 | $-51925.1 \pm 99.8$ | $-51924.9 \pm 99.8$ | 0.004 | 0.0 | $627.8 \pm 20.6$ |
| 0.003 | 0.004 | $-52111.2 \pm 149.0$ | $-52111.0 \pm 148.8$ | 0.004 | 42.8 | $466.9 \pm 13.9$ |
| 0.001 | 0.004 | $-52060.1 \pm 171.8$ | $-52060.0 \pm 171.9$ | 0.004 | 0.0 | $645.6 \pm 19.2$ |

(CELEBA 64)

Table 5: Evaluation of deep Gaussian VAEs (averaged over 5 trials) on real-valued CelebA without any nonlinear preprocessing. Collapse percent gives the percentage of latent dimensions which are within 0.01 KL of the prior for at least 99% of the encoder inputs.

Figure 13: Latent space interpolations from the convolutional VAE trained with KL-Annealing on CelebA.