[Reviews · NeurIPS 2019]

Reviewer 1



Variational Autoencoders (VAEs) are an effective approach to unsupervised learning but they suffer from a problem known as "posterior collapse". There has been a lot of focus on solving this problem from the ML research community. This paper belongs to that line of research. Clarity: The paper is well written however there are few things that hinder clarity. For example having section 5 as is is misleading...it makes it seem as though the same theoretical conclusions made in previous sections also hold for deep VAEs. Furthermore, the discussion makes other misleading statements that were not verified in the draft; for example the statement "We demonstrate empirically that the same optimization issues play a role in deep non-linear VAEs" is misleading because the experiments did not directly show this. Quality: The experiments conducted do not align well with the theoretical analysis in the earlier sections. The experiment section heavily focused on showcasing the performance of KL annealing. The related work section is also limited as there are tons of related work on posterior collapse in VAEs that were not referenced. See for example [1] and [2] below. Significance: I find the paper of limited significance. The analysis for the linear case does not apply to the non-linear case and empirical evidence does not show that the conclusions from the linear case still hold in the nonlinear case. Because of this I think significance is lacking. Minor Comments: 1--Throughout the paper: it's "log marginal likelihood" not "marginal log-likelihood" 2--What happens when the decoder is linear but the encoder is not? Does the analysis on linear decoders still hold? [1] Tackling Over-Pruning in Variational Autoencoders. Yeung et al., 2017. [2] Avoiding Latent Variable Collapse with Generative Skip Models. Dieng et al., 2018.

Reviewer 2



Originality The connection between pPCA and linear VAEs is well-known and already discussed in the literature. However, the paper proposes to analyze the problem of the posterior collapse by inspecting pPCA and linear VAEs. I find this analysis interesting and important. Quite surprisingly, it seems that the posterior collapse follows from the marginal likelihood maximization, and neither the non-linear character of the model nor the ELBO cause this issue. Quality The theoretical results are properly derived and are important for understanding the considered phenomenon. Moreover, the empirical results supports the theoretical analysis. The authors carried out experiments on real-valued MNIST and CelebA datasets. Clarity The paper is clearly written and well organized. The theoretical part is well explained and all concepts are outlined. The only problem with this paper is that after identifying a problem, I would expect a proposition of fixing it. The authors propose to play with this the variance, however, it is quite chaotically described. I would expect a better explanation of their proposition. Significance The problem of the posterior collapse is very important for learning VAEs. The findings of this paper are, thus, of high significance. Remarks: - The "jump" from the theoretical part to experiments is too large. What I mean by that is that a more natural would be to first identify the problem, then propose a solution, and then verify empirically whether the proposed solution is correct. I miss the "solution part" in this paper. - Can we extend the result of this paper to other distributions than Gaussians? As presented in the Appendix, it is not trivial. - The paper considers the continuous latent variables. The conclusion of the paper is that the problem lies in the marginal likelihood optimization. However, taking a discrete-valued latents, is the posterior collapse still the case? ==========AFTER REBUTTAL========== I would like to thank the authors for their rebuttal. I want to inform that I read it as well as I read the other reviews. I am satisfied by the rebuttal and stand by my score. In my opinion the authors can easily address the questions raised by the reviewers in the final version of the paper. I understand some concerns of other reviewers about invalidity of the presented analysis in the case of the non-linear VAEs, however, I still believe that the main result of the paper is interesting. Showing that the problem lies in the marginal log-likelihood rather than the VAE itself is interesting.

Reviewer 3



*** Originality *** This work belongs to the line of theoretical developments of linear autoencoder and proposes to study linear VAEs in the context of posterior collapse. I do not know the area enough to be sure all the work is referenced. However, the related work section is clear and well-written. As a side note, the title can be misleading as the theoretical developments are not enough to "understand" posterior collapse extensively. There might be some other aspects when working with more complex models, mismatch between the true posterior and the approximate posterior etc.. *** Quality *** To my understanding, the analysis of pPCA (which probably has been observed before) and Theorem 1 are rather interesting findings. I appreciate the idea that if a behavior is symptomatic in a simple model (pPCA with Gaussian noise), then it shall not work on more complex scenarios as well. Regarding the experiments, I have a couple remarks: + There is probably no need to point "bad" implementations in this manuscript. However, I never actually saw an open source implementation for a VAE with sigma^2 = 1 and this looks like a bad idea to start with. Can the authors provide more insight on why people would do this in the first place ? + The main experimental results are in Table 1. In particular, they simply suggest that there might be some challenges on initializing the VAEs ? Unfortunately, the theory does not provide much more insight on this ? + Figure 4 suggests that KL annealing might still be useful to "help" the VAE initializing its sigma towards low values. To my understanding, KL annealing was not studied in this manuscript and this is something that was not predicted by the theory ? + "This is strong evidence that the observation noise controls the stability of the stationary points in the non-linear model as in the linear case" -> are two datasets enough to claim strong evidence ? All in all, my concern is that there are some additional claims in the experimental section that might not have a corresponding theory. I think the paper could make it clearer what is expected, or analyzed (in this manuscript or in any other of the related work) and what is not. *** Clarity *** The manuscript is well-written and clear, I appreciated reviewing it. *** Significance *** My main criticism is about the link between the experiments and the theory, which could be made more rigorous or explicit. *** Minor remarks *** 1) Figure 4 and 5 should be permuted 2) Notation for the Encoder E_\psi is not used in the paper 3) Table 1 could have a clearer caption "trained on XX on the training set" ? 4) Figure 4 and 5 are not the easiest to read. Why not indicate to the reader exactly where it is best to see the mass ? ===== After rebuttal ===== The reviewers answere my comments. I still believe there is a mismatch between experiments and theory but this is a good contribution.

[Author Response · NeurIPS 2019]

| Fixed $\sigma^2$ | ELBO | $\sigma^2$-tuned ELBO | Tuned $\sigma^2$ | Posterior collapse (%) | KL Divergence |
|---|---|---|---|---|---|
| 30 | $-1850.0 \pm 29.0$ | $-1374.9 \pm 199.0$ | 4.451 | 91.78 | $10.9 \pm 6.69$ |
| 10 | $-1450.3 \pm 4.17$ | $-1098.2 \pm 28.3$ | 2.797 | 89.08 | $28.8 \pm 1.39$ |
| 3 | $-1114.9 \pm 1.05$ | $-1018.8 \pm 0.99$ | 1.361 | 59.50 | $58.5 \pm 1.39$ |
| 1 | $-1022.1 \pm 5.42$ | $-1018.3 \pm 5.28$ | 1.140 | 8.28 | $125.4 \pm 4.19$ |
| 0.3 | $-1816.7 \pm 270.6$ | $-1104.6 \pm 6.23$ | 1.28 | 1.5 | $179.3 \pm 85.9$ |
| 0.1 | $-3697.3 \pm 493.3$ | $-1190.8 \pm 37.4$ | 0.968 | 1.9 | $368.8 \pm 94.6$ |

Table 1: Evaluation of deep Gaussian VAEs (averaged over 5 trials) on real-valued MNIST. Collapse percent gives the percentage of latent dimensions which are within 0.01 KL of the prior for at least 99% of the encoder inputs. Note that these values differ from those in the current manuscript as we have adopted the procedure from Papamakarios et al. 2017 for ease of comparison (this is ultimately non-linear preprocessing and a constant shift to all values). All results in the paper are now consistent with this.

We are grateful for your comments and suggestions. Our paper provides a thorough, novel, theoretical analysis of the
linear VAE and builds a clear picture of posterior collapse in this model. We showed that the linear VAE can be trained
with ELBO without spurious local maxima and is fully identifiable. Empirically, we explored the extent to which the
linear model can explain observations of the non-linear case. Per reviewer feedback, we have added additional empirical
results (a subset of which are included here) which aim to better explore this relationship.

**Experiments and theory (R1/R3/R4)** We agree that the experiments and theory could have been better aligned. To
correct this we now directly measure posterior collapse statistics in Table 1 (subset included top). The analysis of the
linear model predicts that larger $\sigma^2$ will lead to more posterior collapse and worse ELBO which is generally the case.
However, the non-linear model has some unexplained behaviours — e.g. the best model has some posterior collapse.
**Claims for non-linear case (R1/R4)** We agree that our choice of language could be improved with regards to these
claims and have softened language in these areas of the paper. To help transition between the theory and experiments,
per **R1**'s suggestion, we have clarified that we study the non-linear VAE empirically (and not theoretically) in Section 5.
**Linear decoder with non-linear encoder? (R1)** With a linear decoder and non-linear encoder, Lemma 1 still holds,
and the optimal variational distribution is the same as the true posterior has not changed. However, Corollary 1 and
Theorem 1 no longer hold in general. Even a deep linear encoder will not have a unique global maximum and new
stationary points (possibly maxima) may be introduced to ELBO in general. We have added experiments exploring
different encoders with linear decoders (see Figure 1). We do not expect the linear encoder to be out-performed and
indeed the empirical results support this. Also note that the references you provided do not use Gaussian observation
models and so are much harder to analyze (see Appendix C.1 for an example).
**Related work (R1)** We have included the references and added a long-form related work section to the appendix.
**Significance (R1)**: We acknowledge that the theoretical results apply only to the linear case but argue this is significant
nonetheless. We give a novel interpretation of posterior collapse which is theoretically grounded and opposes existing
folk-wisdom (that the KL term is responsible). With our results, linear VAEs provide a simple, well-understood, test-bed
for analyzing new VAE training strategies. Finally, we proved that linear VAEs are without local maxima and are fully
identifiable (unlike regularized linear autoencoders which only identify the orthogonal subspace [Kunin et al. 2019]).

**Solutions to posterior collapse (R3)** We have identified that
the linear case does not need a novel solution: learning the
observation noise is sufficient for finding the global maximum.
Experimentally, we identified that the non-linear case has chal-
lenges in learning $\sigma^2$ which we hypothesize leads to worse
models. Posterior collapse is a widely studied, challenging
problem which is poorly understood. We believe that the results
in our paper will guide researchers' search for solutions.

Figure 1: VAEs with linear decoders trained on real-valued MNIST. Final average ELBO on training set are (ordered by legend): -1098.2, -1108.7, -1112.1, -1119.6.

**Fixing $\sigma^2$ in the wild (R4)** We believe this stems from a mis-
understanding of VAEs as a regularized autoencoder rather than
a probabilistic model and note it was also observed by Dai &
Wipf, 2019. We can remove this text if preferred.
**Intuitions from Table 1 (R4)** Yes, we have identified that initialization and other pre-processing are critical for training
the non-linear VAEs (which the linear VAE theory does not predict).
**Evidence from two datasets (R4)** We will soften this statement. However, the results around fixed $\sigma^2$ were very
consistent across all experiments: larger $\sigma^2$ learned a less rich representation.

**Additional reference:**  *George Papamakarios, Theo Pavlakou, Iain Murray. Masked Autoregressive Flow for Density*
*Estimation, 2017.*

[Meta-Review · NeurIPS 2019]

There was some disagreement about the merits of this paper amongst the reviewers. The final consensus was that it should be accepted. I ask the authors to be more careful about the extent of claims as pledged to the reviewers in the rebuttal.